# Impact of Insurance Benefits and Education on Point-of-Care Ultrasound Use in a Single Emergency Department: An Interrupted Time Series Analysis

**DOI:** 10.3390/medicina58020217

**Published:** 2022-02-01

**Authors:** Soo-Yeon Kang, Sookyung Park, Ik-Joon Jo, Kyeongman Jeon, Seonwoo Kim, Guntak Lee, Jong-Eun Park, Taerim Kim, Se-Uk Lee, Sung-Yeon Hwang, Won-Chul Cha, Tae-Gun Shin, Hee Yoon

**Affiliations:** 1Samsung Medical Center, Department of Emergency Medicine, Sungkyunkwan University School of Medicine, Seoul 06351, Korea; syrei3.kang@samsung.com (S.-Y.K.); ikjoon.jo@samsung.com (I.-J.J.); guntak.lee@samsung.com (G.L.); jongeun7.park@samsung.com (J.-E.P.); taerimi.kim@samsung.com (T.K.); seuk.lee@samsung.com (S.-U.L.); sygood.hwang@samsung.com (S.-Y.H.); wc.cha@samsung.com (W.-C.C.); taegun.shin@samsung.com (T.-G.S.); 2Department of Emergency Medicine, Graduate School of Kangwon National University, Chuncheon-si 24341, Korea; 3Samsung Medical Center, Department of Nursing, Seoul 06351, Korea; sookyung731.park@samsung.com; 4Division of Pulmonary and Critical Care Medicine, Samsung Medical Center, Sungkyunkwan University School of Medicine, Seoul 06351, Korea; kyeongman.jeon@samsung.com; 5Biomedical Statistics Center, Samsung Medical Center, Research Institute for Future Medicine, Seoul 06351, Korea; swkimid12@naver.com

**Keywords:** point-of-care ultrasound, ultrasonography, emergency department, insurance, education

## Abstract

*Background and Objectives*: Point-of-care ultrasound (POCUS) is a useful tool that helps clinicians properly treat patients in emergency department (ED). This study aimed to evaluate the impact of specific interventions on the use of POCUS in the ED. *Materials and Methods*: This retrospective study used an interrupted time series analysis to assess how interventions changed the use of POCUS in the emergency department of a tertiary medical institute in South Korea from October 2016 to February 2021. We chose two main interventions—expansion of benefit coverage of the National Health Insurance (NHI) for emergency ultrasound (EUS) and annual ultrasound educational workshops. The primary variable was the EUS rate, defined as the number of EUS scans per 1000 eligible patients per month. We compared the level and slope of EUS rates before and after interventions. *Results*: A total of 5188 scanned records were included. Before interventions, the EUS rate had increased gradually. After interventions, except for the first workshop, the EUS rate immediately increased significantly (*p* < 0.05). The difference in the EUS rate according to the expansion of the NHI was estimated to be the largest (*p* < 0.001). However, the change in slope significantly decreased after the third workshop during the coronavirus disease 2019 pandemic (*p* = 0.004). The EUS rate increased significantly in the presence of physicians participating in intensive POCUS training (*p* < 0.001). *Conclusion*: This study found that expansion of insurance coverage for EUS and ultrasound education led to a significant and immediate increase in the use of POCUS, suggesting that POCUS use can be increased by improving education and insurance benefits.

## 1. Introduction

Point-of-care ultrasound (POCUS) is a useful tool that helps clinicians make decisions to properly evaluate and treat patients in the emergency department (ED) [1,2,3,4]. Advances in ultrasound devices, such as portable machines, have improved the accessibility of ultrasound [5,6], and POCUS has become an essential technique for emergency physicians (EPs) for bedside examination of patients [7,8,9]. In South Korea, ultrasound devices were designated as essential equipment for regional and local emergency medical centers by the Emergency Medical Service Act in 2000 [10,11]. In addition, insurance coverage for emergency ultrasound is gradually expanding [11,12], and the use of POCUS in the ED has increased dramatically in recent decades [13].

As the usefulness of POCUS has emerged, several studies have investigated various factors that influence the use of POCUS. Major barriers to its use are related to infrastructure, such as equipment, faculty, and manpower, as well as education, such as ultrasound scanning skills and interpretation. In addition, there are obstacles associated with reporting and billing ultrasound results [14,15,16,17,18]. However, previous studies have been performed through surveys [11,14,15,16,17] or using reimbursement claims data [19,20]. To the best of our knowledge, no study has analyzed the impact of the introduction of specific factors on the use of POCUS in the clinical setting. Therefore, this study aimed to evaluate the impact of some interventions (expansion of benefit coverage of the National Health Insurance (NHI) for emergency ultrasound (EUS) and the introduction of annual ultrasound educational workshops) on the use of POCUS in a single tertiary institute.

## 2. Materials and Methods

### 2.1. Study Design

This was a retrospective study assessing how interventions changed the use of POCUS in the ED of a tertiary academic medical institute in Seoul, South Korea from October 2016 to February 2021. In this study, we chose two main interventions—expansion of benefit coverage of the NHI for EUS and introduction of annual ultrasound educational workshops at the institute. We conducted an interrupted time series analysis (ITS), which provides an estimation of changes in the level and slope among sections of interventions. This study was approved by the Institutional Review Board of Samsung Medical Center (IRB file number 2021-02-158-002).

### 2.2. Study Site

Samsung Medical Center is a local emergency medical center located in an urban area with a 73-bed emergency unit having an annual volume of 70,000 patients and is mainly in charge of treating patients in critical condition. It is an academic medical center with a total of 33 physicians (17 emergency medicine (EM) residents and 16 EPs) and has a 4-year EM residency program. The EUS scan codes were used in October 2016. In the institution, POCUS training for EPs has been strengthened with two attending EPs who have been serving as ultrasonography instructors. As part of POCUS training for EM residents, we ran a 1-month intensive POCUS training program that fully focused on POCUS during weekday working hours in the ED since 2018. In addition, weekly quality assurance (QA) programs for scanned images and annual emergency abdominal ultrasonography educational workshops have been organized. In 2019, the ultrasound fellowship program was launched in the ED to train ultrasound experts.

### 2.3. Intervention

#### 2.3.1. Expansion of Benefit Coverage of the NHI for EUS

The first intervention was insurance benefit coverage for EUS in the emergency and critical care areas in July 2019. South Korea introduced mandatory health insurance (NHI) in 1977 and extended this until it covered the entire population in 1989. The NHI has a well-defined benefits package based on fee-for-service, and Korean medical institutions are partially funded by the NHI only for benefit-coverage services [21,22]. The Ministry of Health and Welfare of South Korea created EUS codes 1 (one-site scanning for diagnosis), 2 (for two-site scanning), and 3 (for complex scanning) in October 2016 nationwide, and billing for EUS was possible. The insurance coverage of EUS was expanded in July 2019 as part of the Korean policy to strengthen health insurance coverage [12]. At this time, there were briefing sessions in the institution and announcements about NHI coverage and claims.

#### 2.3.2. Ultrasonography Workshops

The second intervention consisted of three ultrasonographic workshops. They were held in September 2018, March 2019, and July 2020. The workshop consisted of blended learning through narrated web-based video tutorials prior to hands-on training. All participants viewed three web-based video tutorials at their own pace, covering the extended Focused Assessment for Sonography in Trauma (eFAST), biliary, and reno-urinary scans. Next, each resident attended a 3-h ultrasonography hands-on training session with a standard patient. Then, a brief pathologic image review was performed, and the entire workshop was led by EM ultrasound faculties. First and second year EM residents were required to attend the workshop, and the rest could choose to attend repeatedly. The timeline of the interventions is presented in Appendix A.

### 2.4. Data Collection

Among the patients who visited the ED between October 2016 and February 2021, patients for whom EUS codes were prescribed were screened. First, the patients who received POCUS for procedures such as paracentesis, thoracentesis, and central line insertion, as well as patients without consent regarding the use of their personal medical information, were excluded. In addition, patients with specific chief complaints or Korean Standard Classification of Diseases (KCD) codes not relevant to the POCUS examination, such as needle stick injury or medical certificate issues, were excluded (Appendix A). The EUS rate was set as the number of EUS scans per 1000 eligible patients visiting the ED per month. In addition, the following data of EUS-scanned patients were included: age, sex, Korean Triage and Acuity Scale (KTAS) level at the ER, type of EUS code, and site where POCUS was performed. 

### 2.5. Outcomes

The primary outcome of this study was the change in the level and slope of the EUS rate before and after the expansion of benefit coverage of the NHI for EUS. The secondary outcome was the change in the level and slope of the EUS rate by intervention, including expansion of the NHI and introduction of annual ultrasound workshops. In addition, changes in the EUS rate over time before intervention and according to the presence of physicians participating in intensive POCUS training were evaluated. 

### 2.6. Statistical Analysis

The effects of interventions on POCUS use were evaluated using segmented regression analysis of interrupted time series (ITS) with adjustment for autocorrelation. The number of lags to fit the model was selected using the backward elimination of insignificant lags. This analysis provides an estimation of changes in level immediately after the intervention and the difference between pre-intervention and post-intervention slopes [23]. We compared the level and slope of the segment after the expansion of the NHI and workshops with those of the segment preceding the interventions. The EUS rate per 1000 eligible patients per month did not satisfy normality using the Shapiro-Wilk test; therefore, all analyses were performed with the value of the square root of the EUS rate. In addition, all analyses were adjusted for the presence of residents participating in intensive POCUS programs. Statistical significance was set at *p* < 0.05. Statistical analysis was performed using SAS version 9.4 (SAS Institute, Cary, NC, USA).

## 3. Results

Among the 7462 patients who were prescribed EUS codes during the study period, 5188 patients were eligible for inclusion in the study and were analyzed. We excluded patients who received POCUS for procedures (*n* = 1489), those who did not consent to using the personal medical information (*n* = 526), and those with specific chief complaints or KCD codes not relevant to POCUS examination (*n* = 259) (Figure 1). The mean (standard deviation) patient age was 57 (±19) years, and 55% patients were men. Among them, 65% had KTAS level 3 or higher, and EUS-1 tests of single-site scanning for diagnosis were mainly used. Echocardiography (36%) was the most commonly performed POCUS scan for 5 years, followed by eFAST (17%), biliary tract scan (17%), and lung scan (14%). In addition, abdominal pain (27%), dyspnea (18%), and chest pain (14%) were the most common initial chief complaints (Table 1).

### 3.1. Changes in the EUS Rate after NHI Expansion

The results of the segmented regression analysis of interrupted time series data by NHI expansion are shown in Table 2 and Figure 2. Before the expansion of the NHI, the EUS rate had gradually increased (*p* < 0.001). After the expansion of the NHI, there was a significant change in level such that the EUS rate immediately increased from 15 to 61 (*p* < 0.001). However, the change in slope was significantly decreased (*p* < 0.001). The EUS rate also increased significantly in the presence of physician participation in intensive POCUS training (*p* < 0.001).

### 3.2. Changes in the EUS Rate after Three Ultrasound Workshops and NHI Expansion

The results of the segmented regression analysis of interrupted time series data by four interventions (three workshops and NHI expansion) are shown in Table 3 and Figure 3. The first workshop did not show a significant change in the level and slope of the EUS rate. However, the second workshop, expansion of NHI, and the third workshop were associated with a significant change in the EUS rate. Among them, the difference in the EUS rates before and after NHI expansion was estimated to be the largest. However, after the third workshop, the change in the slope of the EUS rate decreased significantly (*p* = 0.004). In addition, the EUS rate increased monthly until the first workshop (*p* = 0.013) and increased significantly in the presence of physician participation in intensive POCUS training (*p* < 0.001).

## 4. Discussion

POCUS is known to be very useful for primary patient care in the medical field [8,9,24], leading to the conduct of several studies related to its use. However, most were conducted in the form of a survey involving the medical workforce [11,14,15,16,17], without evaluating the impact of some factors on the use of POCUS. Although this study was conducted in a single institute, it is meaningful that ITS analysis was performed on specific factors using five years of actual medical records, and we identified the expansion of insurance coverage and ultrasound education had a significant effect on the immediate increase in the use of POCUS. This result suggests that the use of POCUS might be increased when improving education and insurance coverage, which has been investigated as barrier to the use of ultrasound [15,25,26].

Some studies have identified that the inability to receive ultrasound expenses serves as a barrier to the use of POCUS [14,16]. In addition, some studies have shown that the frequency of ultrasound use has increased since the introduction of reimbursement claims or education on billing practices [19,27,28]. Adhikari et al. identified that educational intervention in the billing process increased the number of POCUS scans billed by 180% [28]. In the institution, as the EUS examination started to be fully covered by NHI in July 2019, there were briefing sessions and announcements about NHI coverage and claims. The expansion of insurance coverage might have allowed emergency physicians to prescribe POCUS without burdening the patient’s medical costs. In addition, it is thought that the emphasis on prescriptions through briefing sessions could increase the EUS rate by reducing prescription omissions after POCUS examination.

Considering that other studies have stated that expansion of benefit coverage from insurance for magnetic resonance imaging encouraged excessive use beyond need [13,29], there may also be concerns that POCUS examinations may have been similarly performed unnecessarily. However, unlike other imaging procedures, an ultrasound in our ED must be performed individually by a physician who determines and orders it to be necessary. Therefore, emergency medical staff cannot afford to perform unnecessary ultrasounds in such an overcrowded environment [16]. Furthermore, the majority of EUS were performed on patients with high severity according to KTAS level, and ultrasounds that could supplement computed tomography images such as echocardiography, lung ultrasound, and eFAST were mainly used; thus, ultrasounds were performed only in necessary medical situations. In addition, the Society of Emergency and Critical Care Imaging has provided recommendations on the desirable use of POCUS in emergency and critical care setting under the Korean Health Insurance System [30]. The EUS was used in accordance with these recommendations, and the indications and outcomes of all ultrasound scans were recorded in the patients’ electronic medical record.

However, the change in slope of the EUS rate after the expansion of NHI, especially after the third workshop, was significantly lower than before. This could be explained by the dramatic positive changes in the EUS rate immediately after these interventions, making the subsequent change appear relatively lower. In addition, the fact that repeated emphasis on prescriptions had not been made might be another reason for the decreased EUS rate, according to a study that stated that timely reminders to physicians about billing helped to increase actual prescription [28].Furthermore, the sharp decrease after July 2020 seems to have had a major impact on the negative change in the overall slope, which coincided with the rapid spread of coronavirus disease 2019 (COVID-19) in South Korea [31,32,33,34]. In Particular, the spread of COVID-19 in Korea was severe from August to September 2020, and from November to February, which correlates with the decline in the EUS rate [31]. Due to concerns about infection control, COVID-19 has forced physicians to minimize direct patient contact and time-requiring tests at the bedside to reduce the risks of exposure for healthcare providers [35,36,37]. In this institution, if a patient is suspected of having COVID-19 due to fever or respiratory symptoms, they must be treated in the febrile respiratory infectious disease unit, which is a negative pressure isolation room outside the ER. However, the unit is not equipped with ultrasound; therefore, physicians must use a hand-held device to perform the ultrasound scan if needed. In addition, it was not easy to apply ultrasound due to problems associated with wearing personal protective equipment before and after the POCUS scan, as well as disinfection of machines. Although many studies have recommended the use of lung ultrasound in the COVID-19 pandemic [38,39,40], the use of ultrasound has significantly reduced in our institution, as reported in other studies [41,42].

As for the EUS rate following the introduction of annual ultrasound workshops, the use of POCUS has increased significantly since the second workshop. Several previous studies have shown an increase in the frequency of POCUS after instituting ultrasound education [43,44]. However, the EUS rate, which increased immediately after each workshop, showed a tendency to decrease after several months in this study. Similarly, the study of 61 physicians by Kim et al. found that the level of ultrasound scan knowledge immediately increased after 5-h ultrasound training but decreased significantly after two months, indicating that POCUS skills could be difficult to maintain for a long period of time [45]. Perhaps, 3-h course workshops in our ED might have been insufficient to maintain these skills for a long time. In addition, there are other barriers that affect POCUS use besides education. One study showed that, the greater the number of physicians in the ED, the more physicians use ultrasound every day [16]. In this study, it seems that the temporary decrease in the number of emergency medicine residents who were preparing for the specialty certification exam from November to January each year also contributed to the decline in the EUS rate. Therefore, the effects of ultrasound workshops alone may not have been long lasting, as barriers to infrastructure, such as ED overcrowding or the lack of manpower, have not been improved. In addition, although there was no significant change in the EUS rate in the first workshop, a significant increase occurred after the second workshop, indicating the importance of repeated training and exposure for ultrasound education. Therefore, additional studies are needed to determine the most effective repetition period and duration for ultrasound education.

There may be concerns that EM residents performed EUS only for the purpose of practicing, rather than for the necessity of examination. However, 65% of patients who received EUS had higher severity than KTAS level 3, compared with about 51% in study [46]. It is suggested that ultrasound would have actually been useful for these critical patients. In addition, looking at the types of ultrasound examination performed, echocardiography and lung ultrasound examinations, which were not included in the ultrasound educational contents, accounted for 50% of the total (Table 1). In addition, analysis of the study participants’ initial chief complaints showed that most of the patients fit the indications for POCUS (Table 1) [47]. This institution is a tertiary hospital located urban area that experiences overcrowding daily. Therefore, rather than considering that our training courses encouraged the practice of unnecessary ultrasound examinations, it is reasonable to note that ultrasounds were actively used for patients in need of differential diagnosis via this high-quality education and training.

As shown in Figure 2 and Figure 3, the EUS rate gradually increased until the first workshop. With the increasing importance of POCUS, this institution has established a sub-specialized section for ultrasound in the ED and has made several efforts to improve the use of POCUS. A 1-month intensive POCUS training program for EM residents was initiated so that the physician could focus on ultrasound scans for a sufficient time without the burden of patient care. Residents screened patients who needed a POCUS examination and scanned under the supervision of faculty who specialize in ultrasonography. It was also mandatory to record the reasons for performing the ultrasound and the results of scan in the patient’s medical records. In addition to ultrasound workshops, the quality assurance (QA) program, which reviews the scanned images every week, began in 2018, allowing EPs to improve their ultrasound interpretation and image acquisition skills. Furthermore, we increased the number of ultrasound machines in our ED and placed them in the patient examination area to improve ultrasound accessibility. As a result of these efforts, the number of ultrasound uses per month increased from less than 30, when the EUS code was first created, to a maximum of 367 per month.

In this study, when a resident participated in intensive training for POCUS, the frequency of POCUS use increased significantly to up to approximately 166 cases per month. It was necessary to adjust the analysis, according to the presence or absence of a resident in charge of POCUS. This reaffirmed the existing study that hypothesized that the use of POCUS may increase when physicians can focus on ultrasound scans without being pressed for time in the ED [16]. During a one-month intensive program, residents were given didactic lectures for abdominal and cardiac ultrasound, performed POCUS on patients, and received direct and indirect formative feedback through direct supervision and a QA program. Although skill maintenance could be an ongoing issue, the experience gained from intensive programs is thought to have facilitated the use of POCUS after program completion. However, how the effects of the widespread use of POCUS, in terms of physicians’ decisions, patient outcomes, and medical costs, is another problem and further research is needed [48].

### Limitations

There are some limitations to our study. First, this was a retrospective study and may not represent actual POCUS scans. There were cases in which POCUS was performed but not prescribed. Eligible patients were analyzed, except for patients with specific chief complaints not related to the POCUS test, but as a result some patients who underwent a POCUS scan were excluded from the analysis. In addition, this study used single institutional data for a specific period; thus, the results cannot be generalized to other settings. The use of ultrasound in patient care may vary according to national and institutional protocols. 

Second, the effect of the expansion of insurance coverage may have been overestimated. Repeated emphasis on prescriptions through briefing sessions and NHI coverage announcements may have reduced prescription omissions, and as a result the number of EUS scans may have increased. In addition, there was no survey of perceptions as to whether the insurance expansion itself motivated physicians to use POCUS; thus, a causal relationship could not be clearly identified.

Third, various factors other than the interventions may have influenced the use of POCUS. Introduction of QA and the 1-month intensive POCUS training programs as well as the increase in the number of ultrasound machines could have also affected the use of POCUS [14,15,16,17]. However, these occurred sequentially around the same time as the first workshop, making it difficult to evaluate the impact of each by ITS analysis. 

Finally, external factors related to ED congestion or infectious disease pandemics may have influenced the results. Physicians may not have been unable to perform POCUS due to ED crowding [49]. However, due to the complexity of the assessment of ED crowding, we could not make any corrections for other factors, except for the number of patients. The study period also included the timeframe spanning the COVID-19 pandemic. This has had a very significant impact on the overall health care environment and is thought to be a factor that counteracts the actual effect of the interventions. However, although the frequency of POCUS use might be disturbed in situations such as pandemic or ED overcrowding, the fact that ultrasounds were performed even in such situations proves that we need a fast, non-invasive POCUS for appropriate patient care.

## 5. Conclusions

This study showed that the expansion of insurance coverage and ultrasound education led to an immediate and significant increase in the use of POCUS, suggesting that POCUS use can be increased by improving education and insurance, which have been reported as barriers. However, the effect of education on the use of POCUS did not appear to be long lasting. Therefore, other factors related to infrastructure should be improved together, and additional studies aimed at determining the optimal interval between effective repetitive education and forming a systematic curriculum are needed.

## Figures and Tables

**Figure 1 medicina-58-00217-f001:**
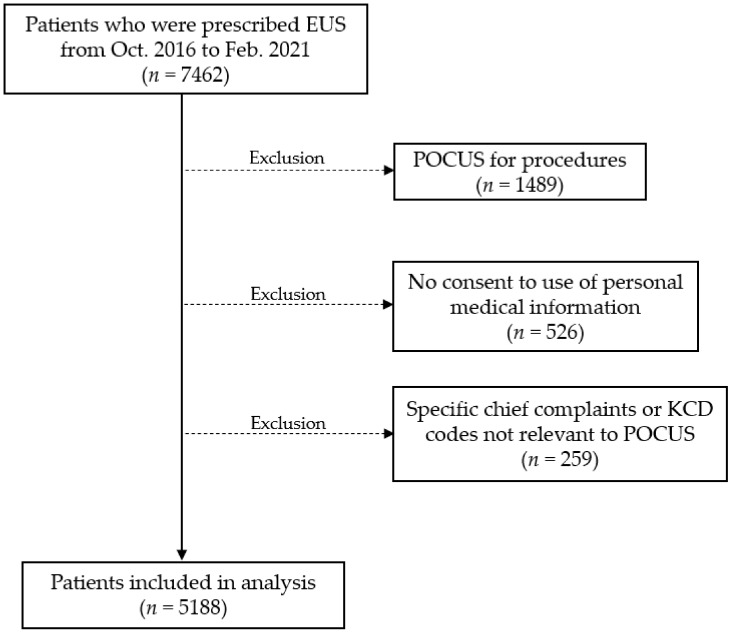
Patient selection. Abbreviations. EUS, emergency ultrasound; POCUS, point-of-care ultrasound; KCD, Korean Standard Classification of Diseases.

**Figure 2 medicina-58-00217-f002:**
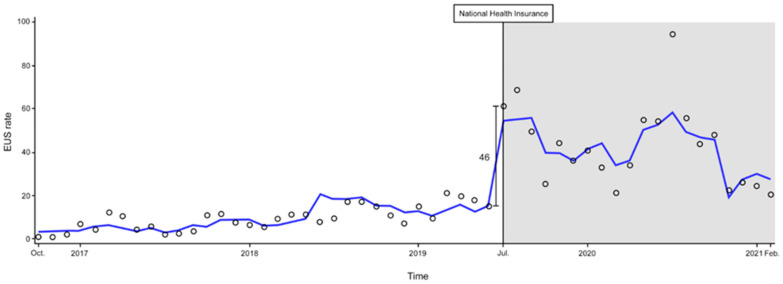
Changes in use of EUS before and after the expansion of National Health Insurance. The monthly EUS rate. Empty round, actual EUS rate; blue line, predicted EUS rate; white background, pre-intervention period; gray background, post-intervention period. Abbreviations. EUS, emergency ultrasound.

**Figure 3 medicina-58-00217-f003:**
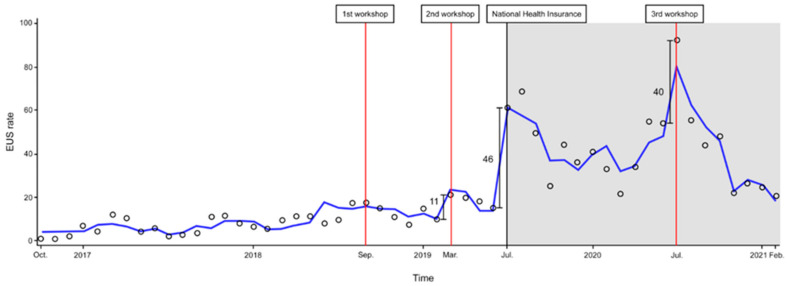
Changes in use of the EUS before and after expansion of National Health Insurance and the ultrasonography workshop. The EUS monthly rate. Empty round, actual EUS rate; blue line, predicted EUS rate; gray background, period after expansion of National Health Insurance; red lines, time point of the ultrasonography workshop interventions. Abbreviations. EUS, emergency ultrasound.

**Table 1 medicina-58-00217-t001:** Baseline characteristics of patients undergoing EUS (*n* = 5188).

Variables	*N* (%)	Variables	*N* (%)
Age, years (mean, SD)	57 (19)	Arrest	53 (1)
Male (%)	2856 (55)	Others	75 (1)
KTAS (%)		Type of EUS	
1 (Resuscitation)	126 (2)	EUS 1	4168 (80)
2 (Emergent)	850 (17)	EUS 2	736 (14)
3 (Urgent)	2391 (46)	EUS 3	284 (6)
4 (Less Urgent)	1743 (34)	Exam Site of POCUS (%) (*n* = 7171)	
5 (Non-Urgent)	78 (1)	Echo	2573 (36)
Initial main chief complaint (%) (*n* = 5188)		eFAST	1229 (17)
Abdominal pain	1422 (27)	Biliary	1191 (17)
Dyspnea	925 (18)	Lung	1037 (14)
Chest pain	698 (14)	Kidney	699 (10)
Trauma	572 (11)	Aorta	176 (2)
Flank pain	452 (8)	GI tract	97 (1)
General weakness/mental change	370 (7)	DVT	80 (1)
Fever	292 (6)	Liver	49 (1)
Shock	172 (3)	Pancreas	40 (1)
Musculoskeletal/Vascular	126/31 (4)		

Abbreviations. EUS, emergency ultrasound; SD, standard deviation; KTAS, Korean Triage and Acuity Scale; POCUS, point-of-care ultrasound; Echo, echocardiography; eFAST, extended focused assessment for sonography in trauma; GI, gastrointestinal; DVT, deep vein thrombosis.

**Table 2 medicina-58-00217-t002:** Effect of expansion of National Health Insurance.

Variables	Regression Coefficient	Standard Error	*p*-Value
Intercept	1.7403	0.1949	<0.001
Time (month)	0.0613	0.0105	<0.001
NHI expansion			
Changes in level	2.8319	0.3412	<0.001
Changes in slope	−0.127	0.0243	<0.001
POCUS physician (presence)	1.3087	0.2621	<0.001
AR4	0.4316	0.1345	0.002

All results were described from the analyses of the square root of the EUS rate and correction of the presence of intensive POCUS training physician (POCUS physician). The Total R-squared value was 0.8897. AR4 is an autoregressive process of the order 4. Abbreviations. NHI, National Health Insurance; POCUS, point-of-care ultrasound; AR, autoregression.

**Table 3 medicina-58-00217-t003:** Effect of introduction of the National Health Insurance and workshops.

Variables	Regression Coefficient	Standard Error	*p*-Value
Intercept	2.005	0.7975	0.016
Time (month)	0.0435	0.0167	0.013
First workshop			
Changes in level	0.0649	0.7307	0.930
Changes in slope	0.0428	0.1878	0.821
Second workshop			
Changes in level	1.8004	0.8539	0.041
Changes in slope	−0.5488	0.355	0.130
NHI expansion			
Changes in level	3.0601	0.6759	<0.001
Changes in slope	0.3895	0.2653	0.150
Third workshop			
Changes in level	1.9066	0.6695	0.007
Changes in slope	−0.3484	0.1135	0.004
POCUS physician (presence)	1.1747	0.2657	<0.001
AR4	0.5242	0.1353	<0.001

All results were described from the analyses of the square root of the EUS rate and correction of the presence of intensive POCUS training physician (POCUS physician). The Total R-squared value was 0.9266. AR4 is an autoregressive process of the order 4. Abbreviations. NHI, National Health Insurance; POCUS, point-of-care ultrasound; AR, autoregression.

## Data Availability

Data related to this study cannot be sent to the outside due to information security policies in the hospital.

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
