# Peer review of "Impact of Insurance Benefits and Education on Point-of-Care Ultrasound Use in a Single Emergency Department: An Interrupted Time Series Analysis"

_medicina, 2022, doi:10.3390/medicina58020217_

Round 1

Reviewer 1 Report

Dear Dr Soo Yeon Kang and authors,

Thank you for sending this interesting manuscript for review.
POCUS is an up and coming modality which is proving to add to the high-value care. Its use is readily increasing globally which makes this a topic of interest. 
As intriguing as this manuscript is, the pattern of the study is more of a quality improvement project than a typical retrospective cohort study. There are multiple interventions which lead to a change in target compliance. These fit the pattern of plan, do, study, act (PDSA) cycles perfectly. Furthermore, the gains post-intervention followed by a dip with time is typical of quality improvement studies.
However, this was not formally planned as a quality improvement project with clear aim of an improvement target predefined. 
Hence, it is difficult to characterize the exact type of study this retrospective data review will fit in. I would suggest to re-look into defining the type of study this retrospective data review fits in. Also, I would suggest to re-run this study in future as a prospective quality improvement project as this would show tangible effect on patient care.
I wish you the best of luck with your study and any future endeavors.

Author Response

Thank you for the valuable point.  I wrote a detailed reply to the attached word file.

Reviewer 2 Report

This is a retrospective study and this always greatly conditions the results.

According to what the authors express, the difference in the use of ultrasound was that it was paid for by insurance. The question is, was there excessive use of its use without need? Was all the increase detected by the authors justified?

If the need for training of doctors is important and on many occasions generating this workshop means that the use of the equipment is due to practice rather than necessity. This point should be made clear in the text.

In figures 2 and 3 there is a drop in the blue line that does not explain well the reason for this drop.

It seems as indicated that the use of ultrasound is not used critically by doctors since according to situations such as emergency saturation its use has been relegated. It seems more than its use is justified because it is a test included in the insurance rather than by the need for its use.

Perhaps a table of the uses of ultrasound in the emergency room would clarify its use a little more and in what cases it was used.

Author Response

Thank you for the valuable points. I wrote a detailed reply to the attached word file.

Round 2

Reviewer 2 Report

Point 1. Thank you for the information and clarification offered. They must reflect this in the article so that it does not arouse controversy with the readers and it is clear that the ultrasound tests are done because they are necessary and performed by the medical team.

Point 2: It is clarified. You can include some sentences similar to the answer so that the formation is completely clear.

Point 3: Thanks for the clarifications.

Point 4: They must point out these limitations in the article. The same answer that they indicate me, more summarized so that the reader has it clear.

Point 5: The inclusion of the table guarantees a better understanding and gives more value to the article.

Author Response

(The authors gave the same response as above.)
